# Tuning Potassium and Magnesium Fertilization of Potato in the South of West Siberia

Vladimir Yakimenko and Natalia Naumova *

Institute of Soil Science and Agrochemistry, Siberian Branch of the Russian Academy of Sciences,
630090 Novosibirsk, Russia; yakimenko@issa-siberia.ru
* Correspondence: naumova@issa-siberia.ru

**Abstract:** Imbalance of nutrients limits crop yields. Although K fertilization receives sufficient attention in research and practice, Mg supply is rather neglected. The effect of Mg fertilization (0, 5 and 10 g Mg/m$^2$), combined with two K fertilization rates (10 and 15 g K/m$^2$), on potato production and soil exchangeable K and Mg was studied in a three-season microplot field experiment in the Novosibirsk region, Russia. Tuber yield did not respond to the increased K fertilization, but increased at 5 and decreased at 10 g Mg/m$^2$. Total Mg concentration in tubers increased at 15 g K/m$^2$, whereas N, P and K were not affected by fertilization. The tuber yield was maximal (3.6 kg/m$^2$) at 10 g K/m$^2$ and 5 g Mg/m$^2$. Soil exchangeable Mg increased by the year, resulting in preferential development of the aboveground phytomass due to apparently increased Mg availability and K/Mg imbalance. Potato production depended on the year, strongly implicating weather conditions. Therefore, the weather and the chemical nature of K and Mg fertilizers (as pertinent to their release mode from fertilizer in soil), are important for balancing their proportions and amounts while assessing interactions among nutrients in potato production and adjusting regional fertilization strategies.

**Keywords:** arable Phaeozem; potato tubers; potato aboveground phytomass; macronutrients; soil exchangeable potassium; soil exchangeable magnesium





## 1. Introduction

Potassium and magnesium participate in a range of essential physiological functions in plants [1,2], thus being indispensable elements of plant mineral nutrition. Total stocks of both elements in the majority of mineral soils are fairly large, primarily depending on soil mineralogy and granulometry, as well as the type of soil formation. The bulk potassium and magnesium contents in agricultural soils of the temperate zone were estimated as averaging 1.5–2.5% [3,4] and 0.5–1.5% [5], respectively. Relatively high bulk K and Mg contents in the major arable soils of Russia, together with fairly regular depth distributions along soil profiles, are often regarded as reason enough not to focus specifically on the elements' content and their changes. However, long ago there appeared reports about increasing K and Mg content depletion in agricultural soils of both light and heavy granulometry [6,7]; this depletion has been markedly increasing under intensive agricultural use [8,9]. This can result in decreased K and Mg concentrations in agricultural produce [10], and consequently, in human diets [11]. In the case of Mg, the latter problem can be exacerbated by the fact that Mg supply during fertilization is often neglected. Therefore, research of the changes in soil K and Mg status under agricultural use are currently continuing and even increasing in prevalence, as the potato is a national staple crop of the country, which is third-ranked in total potato production worldwide [12].

Imbalance of nutrients is one of the main culprits in limiting crop yields and quality [13], including for the potato. Ideally, all interactions of essential plant macro- and micronutrients should be considered while developing fertilization strategies to increase nutrient use efficiency and yield levels [14]. Some earlier studies established antagonistic

interactions between univalent cations, such as $H^+$, $NH_4^+$, and $K^+$ on the one hand, and bivalent cations, such as $Ca^{2+}$ and $Mg^{2+}$, on the other hand, in soil adsorption–desorption processes and in plant uptake from soil and fertilizers [15]. However, some researchers reported either an antagonistic or synergistic relationship between the ions [16,17].

Efficient nutrient management of potato crops is of special importance in Russia and its regions, including West Siberia. The importance of K sources in potato production has been by now very well-documented [18,19]; the potato readily responds to K fertilization in various forms [18,20] of soil, and climatic and agricultural environments [21]. Thus, the importance of potassium fertilizer strategies for achieving high tuber yields and improving the quality of tubers in a sustainable and cost-effective manner is not questioned. In our earlier studies we established the optimal fertilization rate for potato production on Phaeozems [22], one of the most common soil types in agriculturally used forest-steppe lands in the south of West Siberia. However, much less attention has been paid to Mg nutrition [23]; yet, recent meta-analysis of the effects of Mg fertilization on crop yield in various production systems concluded that there is a great potential for Mg management to increase crop yields [24], including the potato. The interaction of nutrients may be drastically influenced by various factors, e.g., environmental conditions, plant species and cultivar. However, the results reported for K and Mg interaction during crop growth and development are not always unambiguous, which can be attributed to differences in soil and climatic conditions between experimental site locations, fertilization rates and biology of the cultivated crops. Since it is important to optimize soil K and Mg regimes in agricultural ecosystems, studies on regional aspects of the issue are warranted. The aim of this study was to assess the effect of Mg fertilization rate gradient, combined with two rates of K fertilization, on potato tuber yield and aboveground phytomass—their chemical element content and changes in plant-available soil K and Mg content in microplot field experiments with potatoes grown on Phaeozem.

## 2. Materials and Methods

### 2.1. Experimental Site and Conditions

The microplot field experiment was conducted at the Field Experiment Station of the Institute of Soil Science and Agrochemistry of the Siberian Branch of the Russian Academy of Sciences (N 54°42′, E 83°15′, Novosibirsk region, Russia). The station is located within a forest-steppe zone with a sharply continental climate; descriptive statistics of the weather conditions of the growing periods in the years when the experiment was conducted (as provided by [25]) are shown in Table 1.

**Table 1.** Some characteristics of the potato-growing periods in the microplot field experiment in the south of West Siberia.

| Property | | 2018 | 2019 | 2020 |
|---|---|---|---|---|
| Duration, days | Total | 87 | 113 | 113 |
| Air temperature, °C | Mean | 19.0 | 17.3 | 18.9 |
| | Min | 4.5 | −5.0 | 1.1 |
| | Max | 33.8 | 33.5 | 36.3 |
| Air temperature aggregate, °C/day | | 1653 | 1955 | 2136 |
| Precipitation, mm | Sum | 184 | 155 | 182 |
| | Max | 19 | 31 | 18 |
| Relative humidity, % | Mean | 74 | 68 | 70 |
| | Min | 15 | 12 | 16 |
| Cloudiness, % | Mean | 52 | 55 | 55 |

Potatoes were grown on loamy arable soil classified as Luvic Greyzemic Phaeozem (Siltic, Aric), according to the World Reference Base for Soil Resources [26]; Phaeozem is the most common soil type in the region. The soil had been in agricultural use for more

than 30 years. Prior to the experiment setup, the soil had the following properties: pH 6.01, 3.2% of soil organic carbon, 10.2 and 7.3 mg/100 g of exchangeable K and Mg in oven-dry soil, respectively (both extracted by 1 M $CH_3COONH_4$ solution).

## 2.2. Experimental Setup

One potato plant (*Solanum tuberosum* L.) cultivar jelly was grown on each microplot sized 0.5 m × 0.5 m. The seed tubers were carefully calibrated (by choosing ones that were *ca.* 70 ± 2% g in mass, elliptic in shape and visually healthy) before placing into soil. Mineral fertilization treatments and rates used in the experiment are described in Table 2. In our earlier studies, fertilization rates of 10 g N, 6 g P and 10 g K per square meter (NPK1 in Table 2) were found to be optimal for potato production on the same soil at the same experimental station [22]. Fertilizers that were added included ammonium nitrate, double superphosphate, potassium chloride and magnesium oxide.

**Table 2.** Mineral fertilization treatments and rates in the microplot field experiment.

| Fertilization Variants | Chemical Element (g/m$^2$) | | | |
|:---:|:---:|:---:|:---:|:---:|
| | N | P | K | Mg |
| No | 0 | 0 | 0 | 0 |
| NP | 10 | 6 | 0 | 0 |
| NPK1Mg0 | 10 | 6 | 10 | 0 |
| NPK1Mg1 | 10 | 6 | 10 | 5 |
| NPK1Mg2 | 10 | 6 | 10 | 10 |
| NPK2Mg0 | 10 | 6 | 15 | 0 |
| NPK2Mg1 | 10 | 6 | 15 | 5 |
| NPK2Mg2 | 10 | 6 | 15 | 10 |

Each microplot was isolated along the entire perimeter by polyethylene film down to 40 cm depth to minimize possible influences from neighbouring microplots and soil. The experiment was performed in a randomized block design with five replications during three consecutive years (2018, 2019 and 2020), with the same microplot location for fertilization treatments. The general view of the experimental site is given in Figure S1.

## 2.3. Phytomass, Soil Sampling and Analyses

Potato plants were harvested on August 31 each year. Tubers were washed, air-dried and weighed to determine the yield. Aboveground phytomass was also sampled and weighed. Then, a representative aliquot was taken from each phytomass (aboveground and tubers) sample and oven-dried (80 °C until a constant mass achieved) to determine dry mass content. Total N, P, K and Mg content analyses were performed after wet acidic digestion of dry phytomass. Briefly, total N was determined by Kjeldahl method; total P was determined calorimetrically (KFK-3KM, Russia), and total K and Mg were measured by flame photometry (PFA-378, Russia) and atomic absorption spectrometry (AAnalyst 200, USA), respectively.

Total removal of K and Mg from the soil by potato phytomass was estimated as the sum of tuber mass multiplied by the element's concentration in tubers, and aboveground phytomass was multiplied by the element's concentration within itself.

Soil was sampled each year at harvest. At each experimental microplot three cores were taken from a 0–20 cm soil layer and bulked together to comprise one composite sample. Field-moist soil samples were sieved through a 2 mm mesh and stored in a refrigerator (+4 °C) before analyses. Soil-available Mg forms were measured by two methods used in routine analyses in Russia: In the first method, ammonium acetate (1 M) extracts were used [27], whereas the second method was based on extraction with calcium chloride (0.0025 M). After extraction, $K^+$ and $Mg^{2+}$ were determined by atomic absorption spectrometry (AAnalyst 400, PerkinElmer, Inc., Waltham, MA, USA).

*2.4. Statistical Analysis*

The data were analyzed by descriptive statistics, ANOVA and GLM (general linear model) analysis, using the Statistica 13.3 software package (TIBCO Software Inc., Palo Alto, CA, USA). Factor effects and mean differences in post-hoc comparisons by Fisher's LSD test were considered statistically significant at the $p \leq 0.05$ level.

## 3. Results

*3.1. Potato Aboveground Phytomass and Tuber Yield*

Since treatments with zero or NP fertilization were also set up and studied for the purpose of validation, rather than for achieving the main aim of the study, the respective results on aboveground phytomass and tuber yield are presented in Table S1. Briefly, averaged over three years, NP and NPK1 fertilization resulted in 35 and 82% higher tuber yields as compared to the non-fertilized variant, respectively. The corresponding increase in aboveground phytomass was more pronounced, reaching 105 and 153% in NP and NPK1 fertilization treatments, respectively, as compared to the non-fertilized variant. Increased rate of K fertilization led to yet higher tuber yield, but only by few percent.

As for the treatments pertaining to the main aim of the study, i.e., interaction between K and Mg fertilization, aboveground phytomass was found to be strongly dependent on the year, and the interaction of the year with fertilization, but showed no response to fertilization alone (Table 3). Tuber yield, with an opposite pattern of yearly changes as compared to the aboveground phytomass (Table 4), revealed some dependence on Mg fertilization as well as the year, albeit the latter was more than four times less compared to the year effect on aboveground potato phytomass. Increased potassium fertilization showed no effect, both on aboveground phytomass and tuber potato production, whereas Mg fertilization at its double rate was found to decrease tuber yield by 1.1–1.2 times per plant (Tables 3 and 4). The interaction factors Mg fertilization × Year and K fertilization × Mg fertilization × Year showed statistically significant, albeit not very marked contributions (5–6% of total variance, Table 3). Notably, at double Mg fertilization rate, the aboveground phytomass increased gradually with each consecutive year, whereas the tuber yield, being lower compared to the yield at the single Mg fertilization rate, was about the same each year (Table S2). The ratio of aboveground phytomass to tuber mass, being on average 0.50, was significantly higher during the third experimental year at the double Mg fertilization rate (0.66 vs. 0.44 and 0.41, $p < 0.0001$).

**Table 3.** ANOVA results: factor contribution (%) to the total variance of potato aboveground phytomass and tubers in the microplot field experiment with potassium (FK) and magnesium fertilization (FMg).

| Factor | Aboveground Phytomass | *p*-Value | Tuber Yield | *p*-Value |
|---|---|---|---|---|
| FK | 0.1 | 0.713 | 0.2 | 0.690 |
| FMg | 0.4 | 0.629 | **12.2** [1] | **0.003** |
| Year | **53.9** | **0.000** | 12.9 | 0.002 |
| FK × FMg | 2.3 | 0.087 | 2.0 | 0.368 |
| FK × Year | 0.2 | 0.792 | 0.2 | 0.885 |
| FMg × Year | **5.1** | **0.032** | 2.8 | 0.581 |
| FK × FMg × Year | **5.6** | **0.021** | 0.3 | 0.991 |
| Error | 32.5 | | 69.6 | |

[1] The contributions of factors that exerted a statistically significant effect ($p \leq 0.05$) and their *p*-values are highlighted in bold.

**Table 4.** Aboveground phytomass and tuber yield (both in g/plant, the same as g/0.25 m²) of potatoes grown for three years in the microplot field experiment with potassium (FK) and magnesium fertilization (FMg).

| Factor [1] | Factor Levels | | | Aboveground Phytomass | | Tuber Yield | |
|---|---|---|---|---|---|---|---|
| | FK | FMg | Year | Mean | ±S.D. | Mean | ±S.D. |
| Total | | | | 393 | ±94 | 810 | ±149 |
| FK | 1 | | | 395 | ±98 | 816 | ±150 |
| FK | 2 | | | 391 | ±91 | 805 | ±148 |
| FMg | | 0 | | 394 | ±89 | 818 b [2] | ±135 |
| FMg | | 1 | | 400 | ±90 | 869 b | ±163 |
| FMg | | 2 | | 385 | ±105 | 744 a | ±122 |
| Year | | | 2018 | 349 a | ±86 | 877 b | ±176 |
| Year | | | 2019 | 340 a | ±59 | 806 a | ±143 |
| Year | | | 2020 | 490 b | ±39 | 748 a | ±87 |
| FK × FMg | 1 | 0 | | 377 | ±87 | 799 a | ±164 |
| FK × FMg | 1 | 1 | | 416 | ±91 | 900 b | ±161 |
| FK × FMg | 1 | 2 | | 393 | ±116 | 749 a | ±76 |
| FK × FMg | 2 | 0 | | 411 | ±90 | 838 | ±99 |
| FK × FMg | 2 | 1 | | 384 | ±88 | 838 | ±165 |
| FK × FMg | 2 | 2 | | 377 | ±98 | 738 | ±157 |
| FK × Year | 1 | | 2018 | 346 a | ±88 | 885 | ±188 |
| FK × Year | 1 | | 2019 | 347 a | ±74 | 819 | ±126 |
| FK × Year | 1 | | 2020 | 493 b | ±37 | 744 | ±95 |
| FK × Year | 2 | | 2018 | 352 a | ±87 | 869 | ±169 |
| FK × Year | 2 | | 2019 | 333 a | ±42 | 793 | ±161 |
| FK × Year | 2 | | 2020 | 486 b | ±42 | 752 | ±81 |
| FMg × Year | | 0 | 2018 | 350 a | ±38 | 894 | ±136 |
| FMg × Year | | 0 | 2019 | 336 a | ±69 | 818 | ±155 |
| FMg × Year | | 0 | 2020 | 496 b | ±40 | 743 | ±60 |
| FMg × Year | | 1 | 2018 | 392 | ±91 | 973 b | ±165 |
| FMg × Year | | 1 | 2019 | 329 | ±65 | 849 b | ±169 |
| FMg × Year | | 1 | 2020 | 478 | ±31 | 786 a | ±99 |
| FMg × Year | | 2 | 2018 | 305 a | ±100 | 766 | ±173 |
| FMg × Year | | 2 | 2019 | 355 a | ±46 | 751 | ±89 |
| FMg × Year | | 2 | 2020 | 495 b | ±46 | 714 | ±89 |

[1] The values for interaction factors of all three main factors are given in Table S2. [2] The differential values for factor levels are highlighted in bold.

Profiles for aboveground potato phytomass and tuber yield obtained using the GLM model, with soil Kex and Mgex as continuous covariates, are shown in Figure 1, visualizing the opposite yearly patterns of aboveground phytomass and tuber yield, and the decreased tuber yield under the doubled rate of Mg fertilizer addition.

### 3.2. The Contents of Macronutrients in the Aboveground Potato Phytomass and Tubers

The total P and Mg concentration in the aboveground potato phytomass was found to be decreased by the increased K fertilization rate (Table 5), whereas N and K concentrations were not affected, as their increase due to higher K fertilization rate was not statistically significant (Table 6). As for the tubers, their total N, P and K concentrations showed no response to any fertilization, but total Mg concentration was 13% higher at the increased K fertilization rate (and proportionally lower in the aboveground phytomass, Table 6). A similar effect was shown by the doubled Mg fertilization rate as compared to the lower and zero rates (Table 6). Both fertilization factors together accounted for 38% of the total data variance.

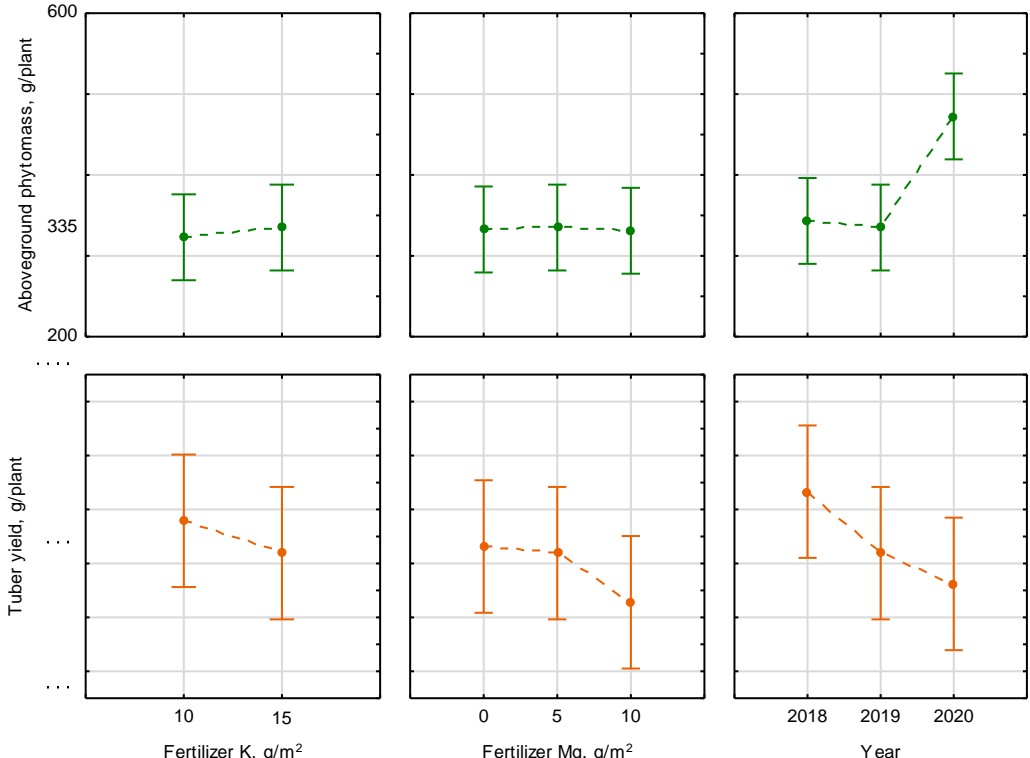

**Figure 1.** Profiles for predicted values of aboveground potato phytomass and tuber yield in the microplot field experiment in the south of West Siberia according to the GLM model ($R^2 = 0.68$ for the aboveground potato phytomass and $R^2 = 0.30$ for the tuber yield).

**Table 5.** ANOVA results: factor contribution (%) to the total variance of potato aboveground phytomass and tubers in the microplot field experiment with potassium (FK) and magnesium fertilization (FMg).

| Factor | N | P | K | Mg |
|---|---|---|---|---|
| | | Aboveground phytomass | | |
| FK | 2.0 | **28.9** [1] | 7.8 | 26.5 |
| FMg | 1.2 | 11.6 | 9.3 | 8.3 |
| FK × FMg | 0.1 | 6.3 | 2.5 | 7.0 |
| Error | 96.6 | 53.6 | 80.4 | 58.1 |
| | | Tubers | | |
| FK | 1.7 | 0.6 | 12.8 | 20.8 |
| FMg | 0.8 | 6.1 | 6.2 | 15.9 |
| FK × FMg | 2.2 | 3.0 | 0.1 | 1.5 |
| Error | 95.3 | 90.4 | 80.8 | 61.7 |

[1] The contributions of factors that exerted a statistically significant effect ($p \leq 0.05$) are highlighted in bold; the contributions of factors with *p*-values in the range $0.05 \leq p \leq 0.10$ are underscored.

Based on total Mg and K concentrations in potato aboveground phytomass and tubers, we estimated the total removal of elements from soil with potato phytomass. Total K removal with potato production did not differ between fertilization treatments, whereas total Mg removal was found to be higher with higher rates of its addition, but only if combined with the low rate of K fertilization (Table 7).

**Table 6.** Total N, P, K and Mg concentrations in aboveground potato phytomass and tubers produced in the last year of the microplot field experiment with potassium (FK) and magnesium fertilization (FMg).

| Factor [1] | Factor Levels | | Element Concentration, % of Dry Mass | | | |
|---|---|---|---|---|---|---|
| | FK | FMg | N | P | K | Mg |
| Aboveground phytomass | | | | | | |
| Total | | | 1.23 | 0.35 | 1.00 | 0.17 |
| FK | 1 | | 1.19 | **0.37 b** [2] | 0.85 | **0.18 b** |
| FK | 2 | | 1.28 | **0.32 a** | 1.14 | **0.16 a** |
| FMg | | 0 | 1.23 | **0.37 b** | 0.82 | 0.16 a |
| FMg | | 1 | 1.28 | 0.34 ab | 0.97 | 0.17 a |
| FMg | | 2 | 1.19 | **0.33 a** | 1.20 | 0.18 b |
| Tubers | | | | | | |
| Total | | | 1.31 | 0.45 | 1.59 | 0.050 |
| FK | 1 | | 1.35 | 0.45 | 1.54 | **0.047 a** |
| FK | 2 | | 1.27 | 0.46 | 1.64 | **0.053 b** |
| FMg | | 0 | 1.34 | 0.44 | 1.63 | **0.048 a** |
| FMg | | 1 | 1.27 | 0.44 | 1.61 | 0.049 ab |
| FMg | | 2 | 1.32 | 0.48 | **1.55** | **0.054** |

[1] Values for the 'FK × FMg' interaction factor are not shown as the factor has a negligible effect (Table 5).
[2] Differential values for factor levels are highlighted in bold at the $p \leq 0.05$ level.

**Table 7.** Total removal of K and Mg (g/plant) by potato phytomass (aboveground phytomass + tubers) at the harvest in 2020 in the microplot field experiment with potassium (FK) and magnesium fertilization (FMg).

| Factor | Factor Levels | | $K_{tot}$ | | $Mg_{tot}$ | |
|---|---|---|---|---|---|---|
| | FK | FMg | Mean | ±S.D. | Mean | ±S.D. |
| Total | | | 3.5 | ±0.6 | 0.21 | ±0.03 |
| FK | 1 | | 3.3 | ±0.6 | 0.21 | ±0.03 |
| FK | 2 | | 3.6 | ±0.6 | 0.20 | ±0.03 |
| FMg | | 0 | 3.4 | ±0.5 | 0.20 | ±0.03 |
| FMg | | 1 | 3.6 | ±0.8 | 0.20 | ±0.02 |
| FMg | | 2 | 3.4 | ±0.5 | 0.22 | ±0.03 |
| FK × FMg | 1 | 0 | 3.2 | ±0.4 | **0.19 a** [1] | **±0.02 a** |
| FK × FMg | 1 | 1 | 3.5 | ±0.9 | 0.21 ab | ±0.02 ab |
| FK × FMg | 1 | 2 | 3.1 | ±0.3 | **0.23 b** | **±0.02 b** |
| FK × FMg | 2 | 0 | 3.6 | ±0.6 | 0.21 | ±0.03 |
| FK × FMg | 2 | 1 | 3.7 | ±0.8 | 0.20 | ±0.02 |
| FK × FMg | 2 | 2 | 3.6 | ±0.6 | 0.20 | ±0.03 |

[1] The values in columns (for a factor) followed by different letters differ at $p \leq 0.05$ level and are highlighted in bold.

### 3.3. Soil Exchangeable K and Mg Content

Exchangeable pools of K and Mg, as expected, had about two thirds of their variance determined by their respective element addition with fertilizer (Table 8): Kex increased 1.3 times as the K fertilization rate increased from 10 to 15 g/m², and Mgex increased proportionally with its addition rate (Table 9, Figure 2). Notably, soil Mgex content showed substantial dependence on the year of study and on the interaction between the year and Mg fertilization (Table 8).



**Table 8.** ANOVA results: factor contribution (%) to the total variance of soil exchangeable K and Mg content in the microplot field experiment with potassium (FK) and magnesium fertilization (FMg).

| Factor | Kex | *p*-Value | Mgex | *p*-Value |
|---|---|---|---|---|
| FK | **63.9** [1] | **0.000** | 0.0 | 0.398 |
| FMg | 0.2 | 0.761 | **65.9** | **0.000** |
| Year | 0.4 | 0.609 | **20.2** | **0.000** |
| FK × FMg | 0.0 | 0.977 | 0.0 | 0.633 |
| FK × Year | **3.1** | **0.030** | 0.0 | 0.480 |
| FMg × Year | 1.2 | 0.577 | **12.1** | **0.000** |
| FK × FMg × Year | 0.3 | 0.960 | 0.0 | 0.946 |
| Error | 30.8 | | 1.8 | |

[1] The contribution of factors that exerted statistically significant effects ($p \leq 0.05$) and their *p*-values are highlighted in bold.

**Table 9.** Soil exchangeable K and Mg contents (mg/100 g o.d.soil) in the microplot field experiment with potatoes with potassium (FK) and magnesium fertilization (FMg).

| Factor [1] | Factor Levels | | | $K_{ex}$ | | $Mg_{ex}$ | |
|---|---|---|---|---|---|---|---|
| | FK | FMg | Y | Mean | ±S.D. | Mean | ±S.D. |
| Total | | | | 16.0 | ±2.4 | 15.1 | ±7.1 |
| FK | 1 | | | **14.1 a** [2] | ±1.3 | 15.0 | ±7.0 |
| FK | 2 | | | **18.0 b** | ±1.6 | 15.1 | ±7.2 |
| FMg | | 0 | | 16.0 | ±2.5 | **7.6 a** | **±0.6** |
| FMg | | 1 | | 16.0 | ±2.5 | **16.1 b** | **±3.7** |
| FMg | | 2 | | 16.2 | ±2.4 | **21.5 c** | **±6.2** |
| Year | | | 2018 | 16.0 | ±2.0 | **10.6 a** | **±2.4** |
| Year | | | 2019 | 15.9 | ±2.6 | **16.8 b** | **±6.9** |
| Year | | | 2020 | 16.2 | ±2.7 | **17.8 c** | **±8.3** |
| FK × FMg | 1 | 0 | | 14.0 | ±0.9 | **7.6 a** | **±0.6** |
| FK × FMg | 1 | 1 | | 14.0 | ±1.1 | **15.9 b** | **±3.6** |
| FK × FMg | 1 | 2 | | 14.3 | ±1.8 | **21.3 c** | **±6.2** |
| FK × FMg | 2 | 0 | | 17.9 | ±1.9 | **7.5 a** | **±0.6** |
| FK × FMg | 2 | 1 | | 17.9 | ±1.8 | **16.3 b** | **±4.0** |
| FK × FMg | 2 | 2 | | 18.1 | ±1.1 | **21.6 c** | **±6.3** |
| FK·Year | 1 | | 2018 | **14.7 b** | **±1.3** | **10.7 a** | **±2.4** |
| FK·Year | 1 | | 2019 | **13.5 a** | **±0.8** | **16.6 b** | **±6.8** |
| FK·Year | 1 | | 2020 | **14.1 ab** | **±1.6** | **17.5 c** | **±8.5** |
| FK·Year | 2 | | 2018 | 17.4 | ±1.7 | **10.5 a** | **±2.5** |
| FK·Year | 2 | | 2019 | 18.2 | ±1.5 | **16.9 b** | **±7.2** |
| FK·Year | 2 | | 2020 | 18.4 | ±1.6 | **18.0 c** | **±8.5** |
| FMg × Year | | 0 | 2018 | 15.4 | ±1.8 | 7.5 | ±0.4 |
| FMg × Year | | 0 | 2019 | 16.0 | ±3.0 | 7.9 | ±0.7 |
| FMg × Year | | 0 | 2020 | 16.5 | ±2.7 | 7.4 | ±0.5 |
| FMg × Year | | 1 | 2018 | 16.1 | ±2.2 | **11.1 a** | **±0.4** |
| FMg × Year | | 1 | 2019 | 15.7 | ±2.6 | **18.4 b** | **±1.0** |
| FMg × Year | | 1 | 2020 | 16.0 | ±2.8 | **18.8 b** | **±1.5** |
| FMg × Year | | 2 | 2018 | 16.5 | ±2.1 | **13.2 a** | **±0.1** |
| FMg × Year | | 2 | 2019 | 15.9 | ±2.5 | **24.1 b** | **±0.6** |
| FMg × Year | | 2 | 2020 | 16.2 | ±2.8 | **27.1 c** | **±2.1** |

[1] Values for the 'FK × FMg' interaction factor are not shown as the factor has a negligible effect (Table 8).
[2] Differential ($p \leq 0.05$) values are highlighted in bold; different letters indicate that the values in a column range (for a factor) differ significantly.

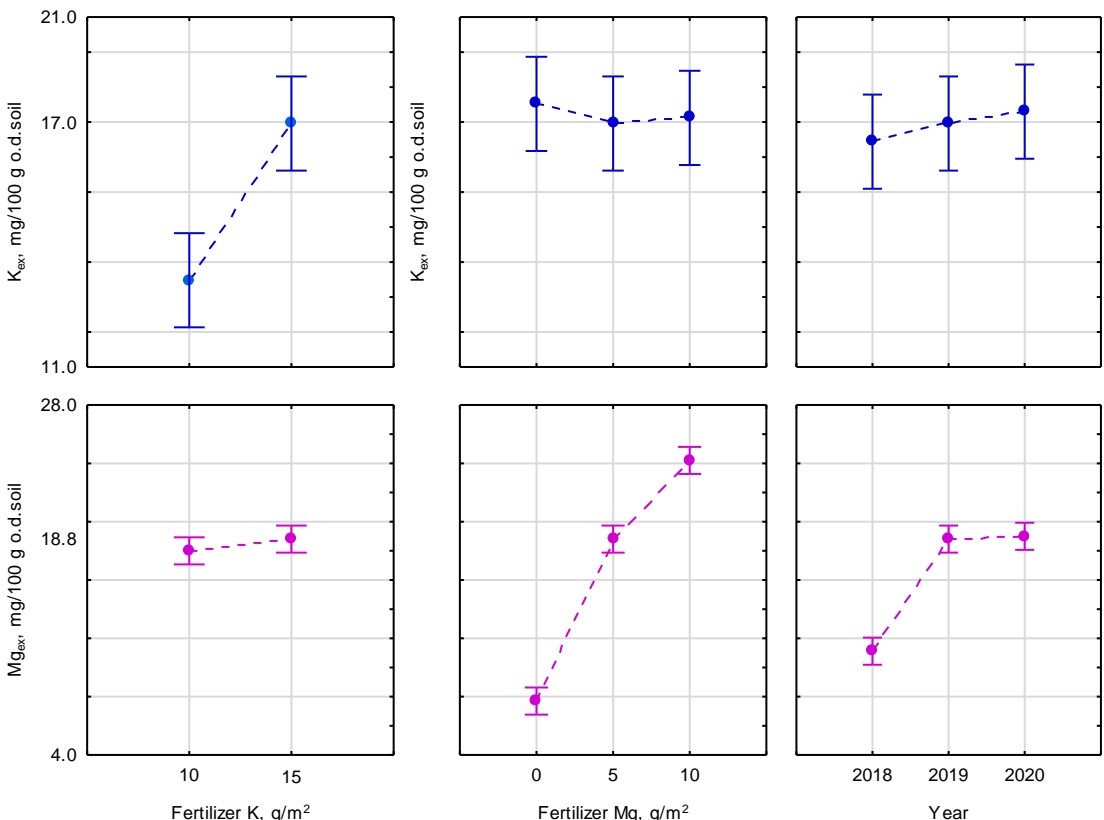

**Figure 2.** Profiles for predicted values of Kex and Mgex according to the GLM model ($R^2$ = 0.69 for Kex and $R^2$ = 0.98 for Mgex). Abbreviations: Kex—soil exchangeable K content, Mgex—soil exchangeable Mg content. Whiskers denote confidence intervals.

At the end of the experiment, exchangeable and readily exchangeable pools of soil K and Mg were measured, and total removal of these elements by potato plants was calculated. The data obtained were used in a general linear model analysis with fertilizer K and Mg addition rates as categorical predictors, and element removal by plants as continuous predictors. Both K and Mg exchangeable pools were strongly dependent on fertilization rates (Figures 3 and 4). Notably, soil exchangeable K, i.e., extracted by ammonium acetate solution, was related to Mg fertilization: without Mg fertilization it was 7.4 mg/100 g soil, as compared to 6.9 ($p$ < 0.005) and 6.6 mg/100 g soil ($p$ < 0.005) at the single and double Mg fertilization rate, respectively. However, soil exchangeable Mg content did not show an association with K fertilization (Figure 4). The removal of soil K and Mg with potato phytomass at harvest (aboveground one tubers) did not affect soil exchangeable K content, but Mg removal was found to decrease readily exchangeable soil K ($p$ = 0.026).

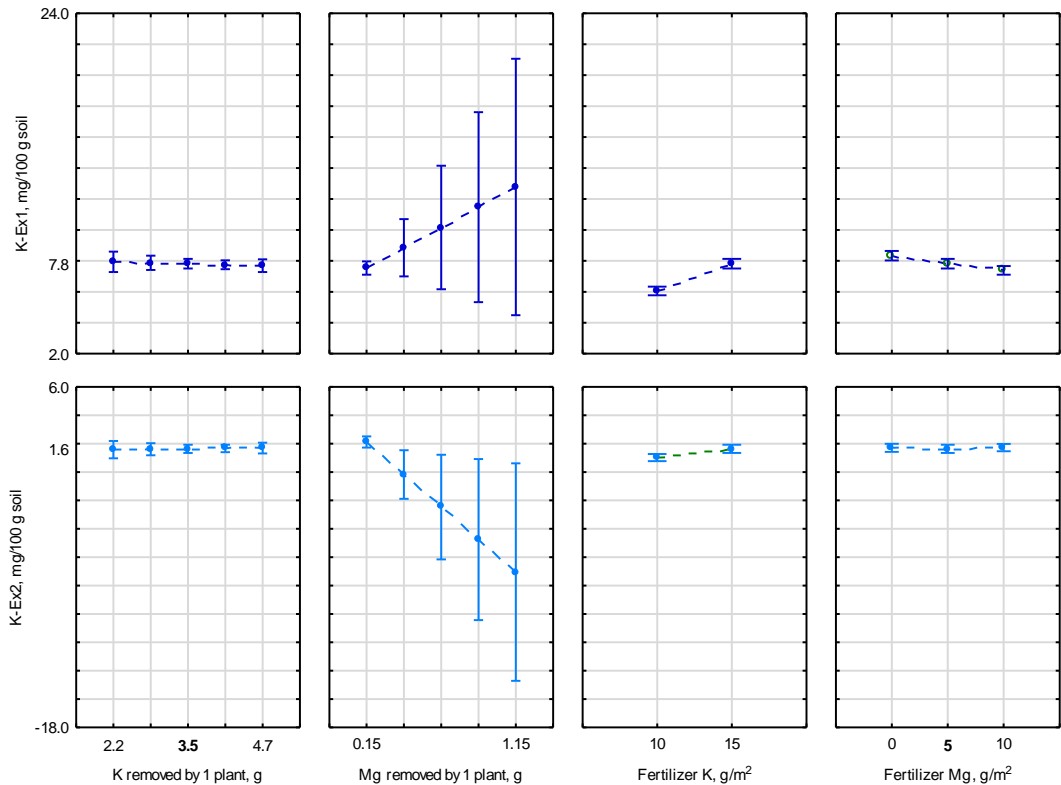

**Figure 3.** Profiles for predicted values of soil exchangeable K according to the GLM model ($R^2 = 0.87$ for K-Ex1 and $R^2 = 0.58$ for K-Ex2). Abbreviations: K-Ex1 and K-Ex2—soil exchangeable K extracted by 1 M $CH_3COONH_4$ and 0.0025 M CaCl2, respectively. Whiskers denote confidence intervals.

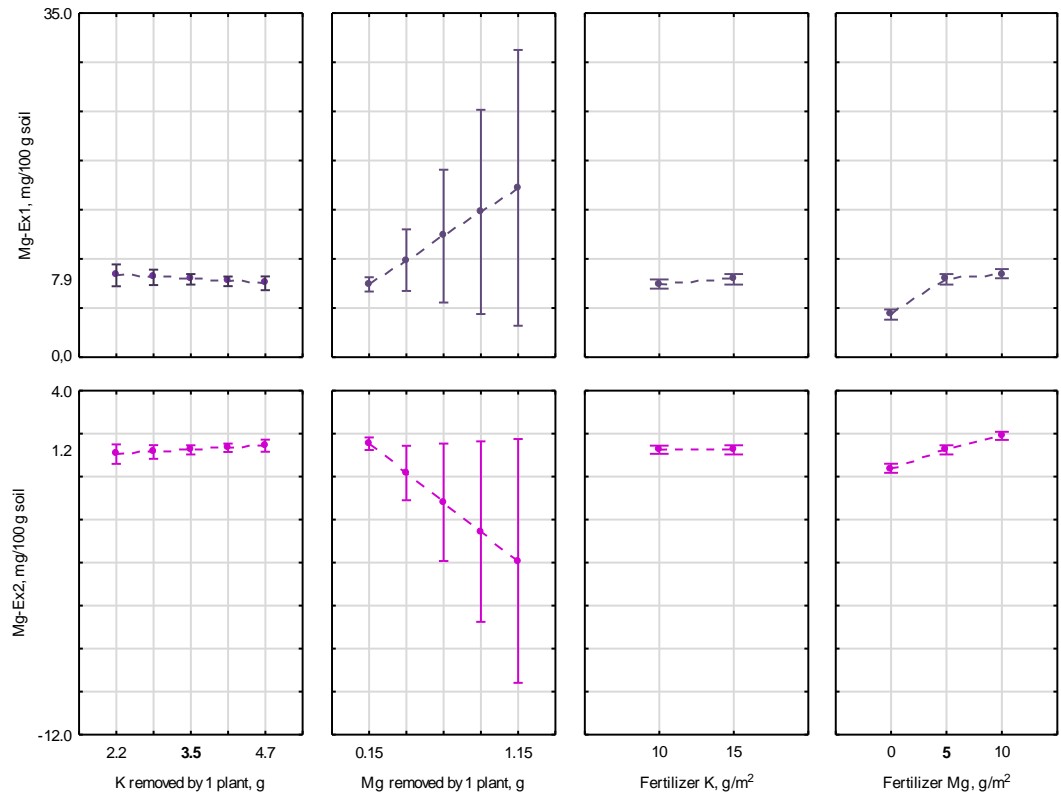

**Figure 4.** Profiles for predicted values of soil exchangeable Mg according to the GLM model ($R^2 = 0.91$ for Mg-Ex1 and $R^2 = 0.85$ for Mg-Ex2). Abbreviations: Mg-Ex1 and Mg-Ex2—soil exchangeable Mg extracted by 1 M $CH_3COONH_4$ and 0.0025 M $CaCl_2$, respectively. Whiskers denote confidence intervals.

## 4. Discussion

### 4.1. K and Mg Fertilization and Potato Yield

Averaged over all treatments and years of our experiment, tuber yield was 3.24 kg/m$^2$ and comparable to values reported for other fertilization experiments with potatoes [28,29], albeit on different soil types (Cambisol), but with some important soil properties, like SOC content, pH etc., being relatively high and fairly similar to the respective properties in our study. The tuber yield in our experiment was higher than in most K fertilization treatments in experiments performed in Egypt [18], an important potato producer.

Some researchers reported increases in potato tuber yield, resulting from increased K fertilization [20,28,29]. Our finding that an increased rate of mineral K fertilization had no effect on potato aboveground phytomass and tuber yield does not agree with such results and strongly suggests that increasing K fertilization rates above those optimal for the area is not economically beneficial for potato production. The same can be concluded about doubled Mg fertilization rate, as the latter resulted in a rather marked decrease in the tuber yield.

The result that Mg fertilization produced maximum tuber yield at a 5 g/m$^2$ fertilization rate, combined with a K rate of 10 g/m$^2$, suggests these fertilization rates are optimal for the soil type (Phaeozem). The increased rates of both nutrients (15 g K/m$^2$ and 10 g Mg/m$^2$) resulted in rather lower tuber yields, i.e., had negative effects, apparently due to the elements' imbalance. Therefore, such fertilization cannot be proposed as an economically reasonable one.

The finding that potato tuber yield was strongly dependent on the year of study seems to implicate weather conditions, confounded with accumulation of poorly soluble MgO, which was applied as fertilizer. Amazingly, despite the fact that the 2018 potato growing period was markedly (26 days) shorter (due to some experiment setting-up logistics) compared to the other two years of the experiment, its weather conditions (Table 1)—with the same average air temperature, narrower temperature fluctuation range, substantial precipitation amount (equaling that of the 26-days longer growing periods in the subsequent two years of the experiment) and higher relative humidity—altogether resulted in the maximum (as compared with 2019 and 2020) tuber yield of 877 g per plant, grown on a microplot of 0.25 m$^2$, which translates to 3.51 kg/m$^2$, or 35.1 t/ha. The fact that the aboveground phytomass produced over the shorter 2018 growing period was the same as the one produced over the much longer period next year also implicates weather conditions as the main driver of potato growth and development under sufficient mineral nutrition, as was the case in our study. Strictly speaking, this finding is not new in itself, but strongly suggests enhanced belowground translocation of photosynthetic assimilates for tuber formation and bulking under an apparently beneficial combination of nutrition and weather situation in 2018. It is noteworthy that, despite the fact that the average cloudiness was about similar in the three growing periods of the study, it was slightly lower in 2018; the difference could have led to the increased, albeit by only a little, amount of photosynthetically active radiation at the plants' development stages, which are crucial for tuber development—resulting in stimulating photosynthesis, product translocation belowground and tuber bulking. Overall, belowground potato production was apparently determined by some weather factor(s), not explicitly accounted for in our experimental fertilization setup. Interestingly, the recently reported results of a potato fertilization study [30] showed a difference (2.76 vs. 3.48 kg/m$^2$) in tuber yield between the two consecutive years of the experiment with rather different meteorological conditions—the year with increased precipitation being beneficial for tuber yield. The difference in tuber yields between these years (1.2 vs. 3.1 kg/m$^2$) was also reported for an experiment conducted in Poland [31]. Thus our results, with an increased tuber yield in 2018 with a higher mean daily precipitation, agree with the cited studies. Moreover, recently reported results from a potato mineral fertilization experiment, performed in another region of Russia [32], showed a very drastic difference between wet and dry years on potato tuber yields (3.6–4.6 vs. 0.9–1.4 kg/m$^2$, respectively). The fact that this decrease occurred in the

control plot with no fertilizers implicates meteorological conditions as the sole driving factor, as in, study weather conditions were confounded by fertilizer application; fertilizers were applied in the year with favorable weather conditions, and the next year, when no fertilizers were applied to study their aftereffect, happened to be meteorologically unfavorable for potato growth and production.

In our study, the result that aboveground potato phytomass was higher in 2020 as compared to 2018, proved that weather conditions in 2020, together with sufficient mineral nutrition, facilitated aboveground plant organ growth, whereas tubers had the opposite yearly pattern. Moreover, the factor of years was combined with accumulation of poorly soluble MgO, especially at double fertilization rate. This apparently increased plant-available Mg content in soil, causing imbalance with potassium, and, in its turn, relatively accelerated aboveground phytomass growth and development. The latter is confirmed by our finding that the ratio of aboveground phytomass to tuber mass was markedly higher during the third experimental year, at the double Mg fertilization rate.

Notably, the maximum tuber yield (900 g per plant, or 3.6 kg/m$^2$) in our experiment was obtained during the shortest growing period, under the addition of 10 and 5 g/m$^2$ of fertilizer K and Mg, respectively. Such tuber yield was 2.1 times higher than Russia's average potato yield for the same year, i.e., 2018 [12]. This result from our study corroborates an earlier conclusion about the optimal fertilizer K rate for the area being 10 g/m$^2$ [22], but primarily draws attention to the need to get a better insight into the interrelationship between seeding dates, potato phenology (in respect to tuber setting) and regional weather pattern changes. In the south of West Siberia, where our experimental site was located, global climate change has prolonged and shifted growing seasons autumn-wise, and increased sums of growing degree days and precipitation [33], which altogether resulted in a tendency for increased plant production across the south of West Siberia in the past several decades. The further anticipated increase in regional plant productivity [34] promotes studies on the growth, development and yield of conventional staple vegetable crops under changed weather patterns during the growing seasons.

Under the highest rates of K and Mg fertilization in our study, Mg concentration in tubers was increased by 13%, averaging 0.41 g Mg/kg of fresh tuber mass. In 2018, the Russian potato supply was estimated at 101 kg/capita/year [33]. The same supply, but with an increased Mg content, as in our experiment, translates on average into Mg intake increase, equivalent to 4% of recommended daily dietary allowance for Mg [35]—which in the long run, especially considering the pattern of decreasing nutritional value of many foods [10], might be beneficial for human health.

Food reward is derived not just from nutrient content, but from sensory qualities as well [36]. As Mg affects plant chlorophyll content and the production and use of carbohydrates—also being involved in the activity of a large number of enzyme systems that are particularly important in the metabolism of carbohydrates—changed Mg content in potato tubers may be associated with changed carbohydrate content. The latter, it its turn, may affect tuber sensory qualities, especially taste. However, we performed neither sensory nor sugar/carbohydrate content assessment of the tubers obtained in our study, as we did not expect small increases in tuber Mg concentration to have any sensory manifestations. Yet, we have currently come to believe that the absence of sensory testing is something of a drawback in our study, and that sensory assessment of foods—especially the staple ones like potato and important vegetable and fruits—should be indispensable in any research involving yields and its properties.

### 4.2. Soil Exchangeable K and Mg Content

The soil exchangeable K content at the start of the experiment (102 mg/kg soil) was on the borderline for inviting K fertilization [37], and increased by 60% on average over all years and K and Mg fertilization treatments. This increase is undoubtedly associated with K application, and it might be safe to implicate that soil cation exchange complexes, as part

of the added fertilizer K, was adsorbed by clay or organic matter surfaces, although it is not possible to discriminate between soil K and fertilizer K.

As for soil exchangeable Mg, its content before the start of the experiment (7.3 mg/kg soil) fell below the range graded as sufficient for crop supply, urging for the use of fertilizer. However, the increase at the end of the experiment, strongly related to Mg application rate and soil exchangeable Mg content, can most likely be attributed to the very poor solubility of MgO when added as fertilizer—retaining a substantial portion in its original form and hence having a cumulative effect on soil exchangeable Mg content.

The fact that soil exchangeable Mg content under both rates of Mg fertilization was substantially lower in 2018, as compared with the other two years, implicates the following factors: (a) Mg uptake by potato plants, (b) weather conditions of the year and (c) the fact that it was the first year of the experiment; addition of poorly soluble MgO each year at the beginning of the growing season most likely had cumulative effect on soil exchangeable Mg [38]. Since in 2018 we did not measure total Mg concentration in potato phytomass, element removal from soil could not be estimated; however, as potato phytomass (aboveground + tubers) was not markedly increased when compared with the phytomass produced during the following two years of the experiment, it could be safe to assume that removal of Mg from the soil by potato plants was not the major cause of decreased soil exchangeable Mg, although the weather, despite a much shorter 2018 growing period, was altogether more beneficial for potato growth and production, and might have stimulated relatively higher Mg uptake by plants. At the same time, although the sum of atmospheric precipitation during the 2018 growing period was close to or the same as that seen in other years, its much shorter duration resulted in a higher mean daily precipitation (2.1 vs. 1.4 mm/day)—most likely enhancing Mg leaching from soil and depleting the exchangeable Mg pool [23].

Interestingly, we found that variations in soil exchangeable K and Mg contents at harvest, i.e., approximately three months after fertilizer addition into soil, was mostly determined by fertilization. As Mg was added in the poorly soluble form of MgO, the finding does not seem surprising, as much of MgO may have remained in the same form in the soil with a not very acidic pH (6.01), and so not be leached or removed by plant production. Moreover, addition of Mg fertilizer in a poorly soluble form can affect soil Mg status for several years after [38]; in our study, where MgO was added every year, its effect on soil exchangeable Mg most likely aggregated. However, K was added as a highly water-soluble KCl that immediately dissolves in soil solution, and then, proceeds further on to sites/zones/agents of its fixation, uptake and leaching. Therefore, after three months when K weathering, translocation and transformation occurred, the strong association of soil K exchangeable content with fertilizer application implicates soil cation exchange complexes in sequestering fertilizer K. We chose poorly soluble MgO as a fertilizer in order not to confound the effect of Mg with the effect of a cation in a highly soluble Mg salt. However, in case of different solubility of chemical compounds added as fertilizers, and hence differential immediate phase distribution of added elements, the effect of fertilizer interactions is more difficult to interpret. Thus, the mode of the fertilizers' release into soil, along with balancing their composition and amount, is important for assessing interactions among nutrients in affecting crop yields.

## 5. Conclusions

Our study showed that potassium fertilization rate, increased above the optimal rate for the area, did not affect potato yield—once more validating the optimal rate. Combined with the latter, the moderate rate of Mg fertilization (5 g/m$^2$) resulted in the highest tuber yield, suggesting that the fertilization scheme is beneficial. The maximum rate (10 g Mg/m$^2$) of Mg fertilization, however, decreased yield significantly as compared to the lower and zero rates, suggesting this combination is counterproductive. The application of fertilizers, including poorly soluble MgO, each year at the beginning of the growing season, resulted in apparently higher levels of plant available Mg at the third consecutive



year of the experiment, causing K/Mg imbalance and preferential development of the aboveground phytomass at the expense of tuber bulking. However, this finding could not be extrapolated for soluble forms of Mg fertilizers because of Mg cation mobility.

We want to emphasize that we see it as a serious drawback that in the current digital age, when all kinds of data sensors, recorders and loggers are widely available, our study was performed without recording yearly dynamics of meteorological conditions at the site, and in particular, precipitation and photosynthetically active radiation. However, our study was far from being exceptional in the field, as the majority of agronomic field experiments performed even very recently, have not employed detailed meteorological records at their field site, to use such data as continuous covariates in analysis of variance to evaluate factor effects—so, we want to encourage other researchers in this direction whenever performing field trials involving plant production.

To conclude, we cannot help but reiterate that "nutrient interactions could be studied for a limited number of crops, nutrients, soil types and climates" (Rietra et al., 2017, p. 1908) [14], and hence other agronomy situations often cannot and even should not benefit from the reported findings. Therefore, studies on the regional aspects of combined K and Mg fertilization effect on potato yield, as well as soil K and Mg status, are yet warranted and will most likely remain such due to climate and related crop phenology changes.

**Supplementary Materials:** The following are available online at https://www.mdpi.com/article/10.3390/agronomy11091877/s1, Figure S1: The general view of the microplot experiment site with potato fertilization on Phaeozem in the south of West Siberia in June, 2020., Table S1: Potato aboveground phytomass and tuber yield (g/m$^2$, mead $\pm$ s.d.) in some mineral fertilization treatments in the microplot field experiment in the south of West Siberia, Table S2: Aboveground phytomass and tuber yield (both in g/plant, the same as g/0.25 m$^2$) of potatoes grown for three years in the microplot field experiment in the south of West Siberia: the data shown are for the levels of the interaction factor of K fertilization (FK) $\times$ Mg fertilization (FMg) $\times$ year.

**Author Contributions:** Conceptualization, V.Y.; methodology, V.Y.; validation, N.N.; formal analysis, N.N.; investigation, V.Y.; resources, V.Y.; data curation, N.N.; writing—original draft preparation, N.N.; writing—review and editing, N.N.; visualization, N.N.; supervision, V.Y.; project administration, V.Y.; funding acquisition, V.Y. All authors have read and agreed to the published version of the manuscript.

**Funding:** This work was supported by The Ministry of Science and Higher Education of the Russian Federation [project number 121031700309-1].

**Institutional Review Board Statement:** Not applicable.

**Informed Consent Statement:** Not applicable.

**Data Availability Statement:** The data presented in this study are available on request from the corresponding author.

**Acknowledgments:** The authors are very thankful to Savenkov O.A. and Barsukov P.A. for their useful comments and suggestions about the manuscript.

**Conflicts of Interest:** The authors declare no conflict of interest. The funders had no role in the design of the study; in the collection, analyses, or interpretation of data; in the writing of the manuscript; or in the decision to publish the results.

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
