# Peer review of "Tuning Potassium and Magnesium Fertilization of Potato in the South of West Siberia"

_agronomy, doi:10.3390/agronomy11091877_

Round 1

Reviewer 1 Report

Manuscript 1353694 by Yakimenko and Naumova reports results of a three-year study of fertilization regimes on potatoes. The study was properly designed and executed. Dissemination of its results will be of benefit to a variety of people involved in potato production. The manuscript is generally well-written, although a few improvements are necessary before publication.

General comments

  1. Several terms used throughout the manuscript needs to be changed: variants to treatments, differential to different, and maximal to maximum.
  2. If possible, the manuscript will benefit from copyediting by a native speaker. It is not essential, but some grammar may use certain improvement.
  3. Figures are blurry. I suggest generating higher-resolution images for printing.
  4. Discussion is a little rambling and occasionally drifts away to discussing weather effects that were not really investigated in the study. I suggest focusing on the study’s goals. What were the effects of K and Mg fertilization, and why were they observed?

Specific comments

Line 90. Solanum, not Solyanum.

Line 91. How was tuber calibration done?

Table 2. Including concentration in a code name will make reading easier. For example, NPK1Mg1 can be changed to NPK10Mg5, and so on.

Lines 109-111. Need to list equipment that was used to perform the tests.

Line 122. Need to indicate the publisher for the software package.

Line 127 and throughout the manuscript. Treatments, not variants.

Table 3. What exactly is “factor contribution”? What do the numbers for aboveground phytomass and tuber yield refer to?

Table 5. The same question as for Table 3.

Lines 195-196. Removal from where?

Lines 273-310. This paragraph is too long. The authors did not specifically test weather effects. Therefore, they should not put too much emphasis on speculating about them.

Lines 322-323. Extrapolating yields from microplots to commercial production is difficult. Furthermore, Russia is a big country with diverse approaches to growing potatoes. Therefore, country-wide averages do not necessarily provide meaningful comparisons.

Lines 346-247. “Did not perform neither” is a double-negative grammatical construction that is not acceptable in English.

Lines 414-422. As far as I understand, studying weather effects was not the main objective of this study. Therefore, this complaint is not really relevant. I suggest deleting this paragraph. It reads as if it is addressed to the authors’ administration as a justification for needing to buy a good weather station. I agree with such a need, but this is not the right place to state it.

Author Response

Point 1

Several terms used throughout the manuscript needs to be changed: variants to treatments, differential to different, and maximal to maximum.

Response 1

We substituted “variants’ with treatments, although the terms are not completely synonymic in some cases the substitution does not work (as in Table 2, for example), and “maximal” with “maximum”. As for “differential”, we believe we used it correctly, meaning “distinctive, discriminating”; besides, the term is often used in the same or similar context in many other MDPI journals, e.g. Life, The Journal of Personalized medicine and others.

Point 2

Figures are blurry. I suggest generating higher-resolution images for printing.

Response 2

We tried to improve the quality by inserting original Statistica graphs.

Point 3

and occasionally drifts away to discussing weather effects that were not really investigated in the study.

Response 3

The weather effects are always implicitly studied in any open field experiments; it is the effect of weather conditions that stipulated the ong-standing requirement for open field agronomy experiments to embrace at least 2-3 years. Therefore we believe it to be quite pertinent to the topic to discuss the effect of the weather conditions and draw attention to the latter once more, however truistic it might seem.  

Point 4

Line 90. Solanum, not Solyanum.

Response 4

Corrected.

Point 5

Line 91. How was tuber calibration done?

Response 5

Explained by adding the text “(by choosing the ones with ca.70±2% g in mass, elliptic in shape and visually healthy)”.

Point 6

Table 2. Including concentration in a code name will make reading easier. For example, NPK1Mg1 can be changed to NPK10Mg5, and so on.

Response 6

Might be the case, but seems a bit too cumbersome. Besides, we do not use the codes often. So we prefer not to change those codes.

Point 7

Lines 109-111. Need to list equipment that was used to perform the tests.

Response 7

The information added.

Point 8

Line 122. Need to indicate the publisher for the software package.

Response 8

Indicated.

Point 9

Line 127 and throughout the manuscript. Treatments, not variants.

Response 9

We substituted “variants’ with treatments, although the terms are not completely synonymic in some cases the substitution does not work (as in Table 2, for example).

Point 10

Tables 3, 5 and 8. What exactly is “factor contribution”? What do the numbers for aboveground phytomass and tuber yield refer to?

Response 10

Factor contribution into the total variance is the percentage of the variance accounted by the factor. We have omitted to indicate %; it is corrected in the revised version.

Point 11

Lines 195-196. Removal from where?

Response 11

Removal from the soil; thus corrected.

Point 12

Lines 273-310. This paragraph is too long. The authors did not specifically test weather effects. Therefore, they should not put too much emphasis on speculating about them.

Response 12

Specifically not, but implicitly yes, as in all several years long filed experiments; besides, the paragraphs shows that one has to really dig in the body of the published matter to find the needed information, as weather conditions are most of the times tested implicitly, rather than specifically.

Point 13

Lines 322-323. Extrapolating yields from microplots to commercial production is difficult. Furthermore, Russia is a big country with diverse approaches to growing potatoes. Therefore, country-wide averages do not necessarily provide meaningful comparisons.

Response 13

We absolutely agree with you in that the country is big, but believe that comparisons should be done to provide some kind of a general view.

Point 14

Lines 346-347. “Did not perform neither” is a double-negative grammatical construction that is not acceptable in English.

Response 14

Of course; sorry for being hasty. We corrected it in the revised version.

Point 15

Lines 414-422. As far as I understand, studying weather effects was not the main objective of this study. Therefore, this complaint is not really relevant. I suggest deleting this paragraph. It reads as if it is addressed to the authors’ administration as a justification for needing to buy a good weather station. I agree with such a need, but this is not the right place to state it.

Response 15

We are sorry, but it is not a complaint; the main aim of the paragraph is to draw attention of other researchers in this direction whenever performing field trials involving plant production. The paragraph totally falls within the Conclusion genre, so we would like not to remove it.

Reviewer 2 Report

The paper „When more produces less: tuning potassium and magnesium fertilization of potato in the south of West Siberia” describe the effect of  Mg fertilization rate gradient, combined with  two rates of K fertilization, on potato tuber yield and aboveground phytomass, their chemical element content, as well as changes in plant available soil K and Mg content in the microplot field experiment with potato grown on Phaeozem. The subject of the article is consistent with the scope of the Agronomy journal. The specific comment are listed below:

  1. The manuscript was prepared carefully, however some editorial errors should be corrected (for example see line 47).
  2. The Figure 1,2, 3,4 are indistinct. The authors should improve the quality of this figure. Moreover, the data showed on this figure is not clear. The axis should be described. Maybe, you can show this data more clearly.

Author Response

Point 1

The manuscript was prepared carefully, however some editorial errors should be corrected (for example see line 47).

Response 1

Thank you very much! We corrected the text in the indicated line, and checked the manuscript for typos and such.

Point 2

The Figure 1,2, 3,4 are indistinct. The authors should improve the quality of this figure. Moreover, the data showed on this figure is not clear. The axis should be described. Maybe, you can show this data more clearly.

Response 2

The graphs were prepared in Statistica v.13.3 with default settings, therefore the axis titles were on the above and on the left of the graphs. So we changed the titles’ positions to the conventional ones, i.e. on the right and below of the graphs. As for the quality, we inserted the original Statistica graphs as they seem more distinct. 

Reviewer 3 Report

The article entitled “When more produces less: tuning potassium and magnesium fertilization of potato in the south of West Siberia" presents the results of a study which aimed to determine the effect of Mg fertilization (0, 50 and 100 kg Mg/ha), combined with two K fertilization rates (100 and 150 kg K/ha), on potato production and soil exchangeable K and Mg content. The research was conducted in the Novosibirsk region, Russia

The studies presented are of rather regional importance. In the world literature, the response of potatoes to fertilization with potassium and magnesium has been known for many years. Thus, the conducted research does not bring any news to the discipline of agriculture. Even the authors themselves notice it (in their conclusions), although they state that this type of research is still up-to-date due to the changing climatic conditions.

Specific comments

In my opinion, the title of the article should be changed.

Lines 12-13: there is: “Increased K fertilization showed no effect on tuber yield, 12 whereas the latter increased at 50 and decreased at 100 kg Mg/ha” - I do not understand this sentence - needs improvement.

Line 15: there is: “(3.6 kg/m2 ) at 100 kg K/ha and 50 kg Mg/ha´- the authors use different units - once they convert the data into ha, once per m2 - this must be standardized.

Line 85: The SOC abbreviation appears for the first time - please explain.

Line 85: the „o.d.” abbreviation appears for the first time - please explain

The Experimental setup section must be improved. The authors explained what doses were used on NPK1 objects, while it is not known what is No, NP, NPK2, Mg1, and Mg2 (data from Table 2). Again the authors use different units - once they convert the data into ha, once per m2 - this must be standardized.

Table 6 – In my opinion the authors should add different letters indicate that the  values in a column range (factors) differ significantly as in Table 4.

In the methodology, the authors should provide the formula as they calculated the total removal of K and Mg (g / plant) by potato phytomass. The information on lines 195-196 is insufficient

In Figures 1 and 2 a description of the x axis should be added. What does 10, 14 (?) and 0, 5, 10 mean - what is the unit? Without description, the figures are illegible and incomprehensible.

Why the doses of K in figure 3 are 10, 15 and in figures 1 and 2 - 10, 14?

I have no comments on the discussion part.

Author Response

Point 1

The studies presented are of rather regional importance. In the world literature, the response of potatoes to fertilization with potassium and magnesium has been known for many years. Thus, the conducted research does not bring any news to the discipline of agriculture. Even the authors themselves notice it (in their conclusions), although they state that this type of research is still up-to-date due to the changing climatic conditions.

Response 1

We agree that our study is mostly of regional importance, but the same is true for much of the agronomy research in the open field. At the same time we wanted our results detailing the Mg fertilization gradient on potato to be a contribution to the global view /meta analysis  on potato production, since the country’s and the region’s research results on this topic are scarcely, if at all, represented in reviews. 

Point 2

In my opinion, the title of the article should be changed.

Response 2

We do not mind to change the title, but did not manage to come up with something very different (alas) and did not want it to be cumbersome.

Point 3

Lines 12-13: there is: “Increased K fertilization showed no effect on tuber yield, whereas the latter increased at 50 and decreased at 100 kg Mg/ha” - I do not understand this sentence - needs improvement.

Response 3

We tried to improve the sentence as far as we had been able to understand the point.

Point 4

Line 15: there is: “(3.6 kg/m2 ) at 100 kg K/ha and 50 kg Mg/ha´- the authors use different units - once they convert the data into ha, once per m2 - this must be standardized.

Response 4

We absolutely agree; since it was a microplot experiment, it is more correct to present all the rates as per the meter squared, and so we did in the revised version of the manuscript.

Point 5

Line 85: The SOC abbreviation appears for the first time - please explain.

Response 5

Explained.

Point 6

Line 85: the „o.d.” abbreviation appears for the first time - please explain

Response 6

Explained.

Point 7

The Experimental setup section must be improved. The authors explained what doses were used on NPK1 objects, while it is not known what is No, NP, NPK2, Mg1, and Mg2 (data from Table 2). Again the authors use different units - once they convert the data into ha, once per m2 - this must be standardized.

Response 7

We are afraid, we could not understand the point, as all the rates are explicitly indicated in Table 2; therefore we could not address the issue. We want to reiterate that we saw no point in presenting and discussing all the variants in the manuscript, since the response of potatoes to fertilization with NP and NPK has been known for many years (though it does not mean that they are universally similar in size etc). 

The units were standardized (see response 4).

Point 8

Table 6 – In my opinion the authors should add different letters indicate that the values in a column range (factors) differ significantly as in Table 4.

Response 8

Letters added.

Point 9

In the methodology, the authors should provide the formula as they calculated the total removal of K and Mg (g/plant) by potato phytomass. The information on lines 195-196 is insufficient

Response 9

The required information was added in the M&M section.

Point 10

In Figures 1 and 2 a description of the x axis should be added. What does 10, 14 (?) and 0, 5, 10 mean - what is the unit? Without description, the figures are illegible and incomprehensible.

Response 10

We are very sorry for the unconventional placement (actually, just default by Statistica) of the axes titles above (instead of below) and on the right (instead of the left) of the axes. Obviously such unconventional placement turned out to be misleading, so we substituted the titles’ placement with the conventional one.

Point 11

Why the doses of K in figure 3 are 10, 15 and in figures 1 and 2 - 10, 14?

Response 11

It was a typo overlooked in the data file used to create those graphs, thank you for your attentiveness. Corrected in the revised version.

Round 2

Reviewer 3 Report

I thank the authors for responding to my comments. I still think that the presented results are not innovative, although the article has been improved. Most of my comments have been included by the authors in the new version of the manusccript.